# PlayFusion: Skill Acquisition via Diffusion from Language-Annotated Play

**Abstract:** Learning from unstructured and uncurated data has become the dominant paradigm for generative approaches in language and vision. Such unstructured and unguided behavior data, commonly known as *play*, is also easier to collect in robotics but much more difficult to learn from due to its inherently multimodal, noisy, and suboptimal nature. In this paper, we study this problem of learning goal-directed skill policies from unstructured play data which is labeled with language in hindsight. Specifically, we leverage advances in diffusion models to learn a multi-task diffusion model to extract robotic skills from play data. Using a conditional denoising diffusion process in the space of states and actions, we can gracefully handle the complexity and multimodality of play data and generate diverse and interesting robot behaviors. To make diffusion models more useful for skill learning, we encourage robotic agents to acquire a vocabulary of skills by introducing discrete bottlenecks into the conditional behavior generation process. In our experiments, we demonstrate the effectiveness of our approach across a wide variety of environments in both simulation and the real world. Video results available at https://play-fusion.github.io.

**Keywords:** Diffusion Models, Learning from Play, Language-Driven Robotics

## 1 Introduction

Humans reuse past experience via a broad repertoire of skills learned through experience that allows us to quickly solve new tasks and adapt across environments. For example, if one knows how to operate and load a dishwasher, many of the skills (e.g., opening the articulated door, adjusting the rack, putting objects in) will transfer seamlessly. How to learn such skills for robots and from what kind is a long-standing research question. Robotic skill abstraction has been studied as a way to transfer knowledge between environments and tasks [1, 2, 3]. It has been common to use primitives as actions in the options framework [4, 5], which are often hand-engineered [6, 7, 8, 9, 10, 11] or learned via imitation [12, 13, 14]. These allow for much more sample-efficient control but require knowledge of the task and need to be tuned for new settings. On the other hand, there have been efforts to *automatically* discover skills using latent variable models [15, 16, 17, 18, 19, 20, 21, 22]. While they can work in any setting, such models are often extremely data-hungry and have difficulty scaling to the real world due to the data quality at hand.

As a result, real-world paradigms are based on imitation or offline reinforcement learning (RL) but both these require several assumptions about the datasets. In imitation learning, human teleoperators must perform tasks near-perfectly, reset the robot to some initial state, perform the task near-perfectly again, and repeat several times. In offline RL, data is assumed to contain reward labels, which is impractical in many real-world setups where reward engineering is cumbersome. In contrast, it is much easier to collect uncurated data from human teleoperators if they are instructed only to explore, resulting in *play data* [21, 22, 23]. Learning from play (LfP) has emerged as a viable alternative to traditional data collection methods for behavior generation. It offers several advantages: (1) it is

Submitted to the 7th Conference on Robot Learning (CoRL 2023). Do not distribute.

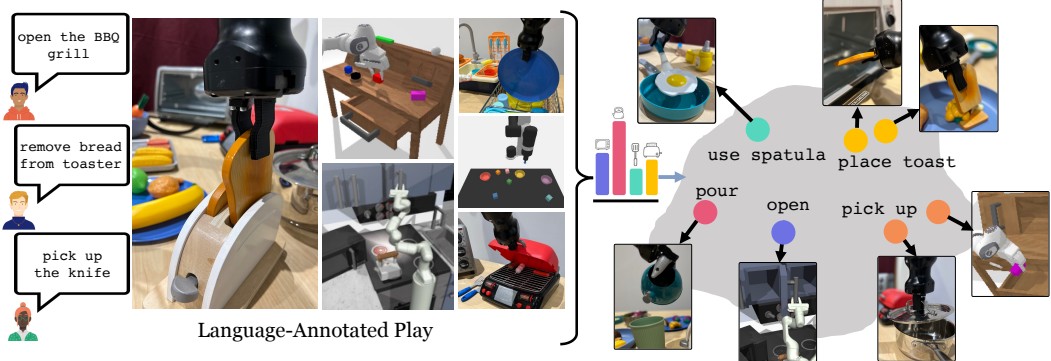

Figure 1: Across multiple real-world and simulated robotic settings, we show that our model can extract semantically meaningful skills from language-annotated play data. Such data is highly multimodal and offers no optimality guarantees. Video results of PlayFusion are available at https://play-fusion.github.io.

efficient because large datasets of play can be collected without the need for setting up and executing perfect demonstrations, and (2) the data collected is rich and diverse because it contains a broad distribution of behavior ranging from completions of complex tasks to random meandering around the environment. An important quality of such data is that it is grounded with some semantic goal that the "player" is aiming to achieve. We believe a simple abstraction for this is *language* instructions, which can describe almost any play trajectory.

A major challenge in learning from play is that the data is highly multimodal, i.e., there are many different ways to achieve a specific goal, and given a sample from the play data, there are many different goals that could have generated it. One popular way to handle highly multimodal data is by modeling the full distribution via generative models. In recent years, there has been remarkable progress in large generative models [24, 25, 26, 27], especially in the class of diffusion models [28, 29], which have been shown to generate high-resolution images – a property well suited for vision-based robotic control. In fact, diffusion models have shown to be effective in capturing complex, continuous actions [30, 31, 29, 32, 33] in the context of robotics. However, these diffusion model-based approaches have not been empirically shown yet to work on unstructured data. We argue that the ability of diffusion models to fully capture complex data paired with their potential for text-driven generation can make them good candidates to learn from language-annotated play data.

One additional consideration is that in reality, humans only deal with a few skills. Almost every task manipulation task involves some grasping and some post-grasp movement. We believe that learning *discrete skills* will not only make the whole process more efficient but will also allow interpolation between skills and generalizations to new tasks. To address this, we propose **PlayFusion**, a diffusion model which can learn from language-annotated play data via discrete bottlenecks. We maintain the multimodal properties of our current system while allowing for a more discrete representation of skills. Empirically, we show that our method outperforms state-of-the-art approaches on six different environments: three challenging real-world manipulation settings as well as the CALVIN [34], Franka Kitchen [22], and Ravens [35, 36] simulation benchmarks.

## 2 Related Work

**Goal and Language Conditioned Skill Learning**   One method of specifying the task is via goal-conditioned learning, often by using the actual achieved last state as the goal [37, 38, 39, 40]. There is also recent work on using rewards to condition robot behavior [41], but this requires a reward-labeled dataset, which makes stronger assumptions than play data. Furthermore, there is a large body of work on language-conditioned learning [42, 36, 43, 44, 45, 46, 47], which specifies the task through language instructions. Instead of conditioning the policy on fully labeled and curated data, we take advantage of unstructured play data which is annotated with language in hindsight.

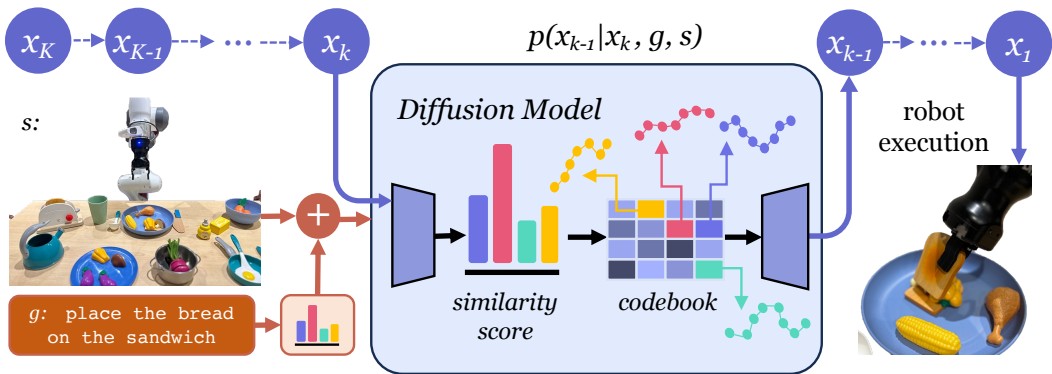

Figure 2: Overview of how PlayFusion extracts useful skills from language-annotated play by leveraging discrete bottlenecks in both the language embedding and diffusion model U-Net. We generate robot trajectories via an iterative denoising process conditioned on language and current state.

**Learning from Play**   Unlike demonstrations, play data is not assumed to be optimal for any specific task as it is collected by human teleoperators who are instructed only to explore. Play-LMP and MCIL [21, 48] generate behaviors by learning motor primitives from play data using a VAE [49, 50]. RIL [22] is a hierarchical imitation learning approach and C-BeT [23] generates behaviors using a transformer-based policy and leverages action discretization to handle multimodality. LAD [51] incorporates diffusion for learning from play, but keeps several components of VAE-based approaches for encoding latent plans; we forgo those elements completely.

**Behavior Modeling with Generative Models**   A promising architecture for behavior modeling with generative models is the diffusion probabilistic model [28, 52, 53, 54]. Diffuser [30], Decision Diffuser [31], Diffusion-QL [29] and IDQL [32] apply diffusion models to the offline reinforcement learning (RL) problem. In real-world robotic applications, Diffusion Policy [33] demonstrated strong results in visuomotor policy learning from demonstrations. Different from these works, we learn from play data containing semantic labels instead of offline RL datasets or expert demonstrations. Some approaches [55, 56] incorporate diffusion in robotics but not for generating low-level actions.

**Discrete control**   A key challenge in robot learning is the exponentially large, continuous action space. Option or skill-based learning is appealing as it can circumvent this problem and allow the agent to learn in a structured, countable action space [57, 58, 59, 60]. Learned action discretization [52, 23] has allowed approaches to scale to complex tasks. C-BeT [23] applies real-world robotic control with transformers [41, 61, 62] to the goal-conditioned setting; [63] train a dynamics model over discrete latent states. We leverage the discrete properties of VQ-VAEs and their natural connection to language-labeled skills.

## 3   Background

**Denoising Diffusion Probabilistic Models (DDPMs)**   DDPMs [28] model the output generation process as a denoising process, which is often referred to as Stochastic Langevin Dynamics. To generate the output, the DDPM starts by sampling $x^K$ from a Gaussian noise distribution. It then performs a series of denoising iterations, totaling $K$ iterations, to generate a sequence of intermediate outputs, $x^k, x^{k-1}, \cdots, x^0$. This iterative process continues until a noise-free output $x^0$ is produced. The denoising process is governed by the following equation:

$$x^{k-1} = \alpha(x^k - \gamma\epsilon_\theta(x^k, k) + N(0, \sigma^2 I)) \tag{1}$$

Here, $\epsilon_\theta$ represents the noise prediction network with a learnable parameter $\theta$, and $N(0, \sigma^2 I))$ denotes the Gaussian noise added at each iteration. This equation is used to generate intermediate outputs with gradually decreasing noise levels until a noise-free output is obtained. To train the DDPM, the process begins by randomly selecting $x^0$ from the training dataset. For each selected sample, a

denoising iteration $k$ is randomly chosen, and a noise $\epsilon^k$ is sampled with the appropriate variance for the selected iteration. The noise prediction network is then trained to predict the noise by minimizing:

$$\mathcal{L} = ||\epsilon^k - \epsilon_\theta(x^0 + \epsilon^k, k)||^2 \tag{2}$$

**Discrete Representations** We utilize VQ-VAE [64] inspired models in PlayFusion as they can provide a way to discretize the skill space. Given an input $x$, a VQ-VAE trains an encoder $E$ to predict latent $E(x) = z$ and maintains a codebook of discrete latent codes $e$. The VQ layer selects $j$ as $\arg\min_i ||z - e_i||$, finding the closest code to the embedding, which is used to reconstruct $x$. The training loss is

$$\mathcal{L}_{\text{VQVAE}} = \mathcal{L}_{\text{recon}}(x, D(e_j)) + ||z - sg(e_j)||_2 + ||sg(z) - e_j||_2 \tag{3}$$

where $D$ is the VQ-VAE decoder. The reconstruction loss is augmented with a quantization loss, bringing chosen codebook embedding vectors $e_j$ toward the encoder outputs in order to train the codebook, as well a loss to encourage the encoder to "commit" to one of the embeddings.

**Learning from Play Data (LfP)** In the LfP setting, we are given a dataset $\{(s,a)\} \in S \times A$. There are no assumptions about tasks performed in these sequences or the optimality of the data collection method. Similar to the formulation of [23], the goal is to learn a policy $\pi = S \times S \to A$ where the input is the current state $s_t$ and goal $g = s_T$. In some cases, (including ours), the goals are instead described via language annotations.

## 4   PlayFusion: Discrete Diffusion for Language-Annotated Play

Humans do not think about low-level control when performing everyday tasks. Our understanding of skills like door opening or picking up objects has already been grounded in countless prior experiences, and we can comfortably perform these in new settings. Skills are acquired through our prior experiences – successes, failures, and everything in between. PlayFusion focuses on learning these skills through *language-annotated* play data.

However, learning from play data is still difficult as continuous control skills are not easy to identify due to several challenges: (1) data can come from multiple modalities as there are many actions that the robot could have taken at any point, (2) we want the model to acquire a vocabulary of meaningful skillsm and (3) we want to generalize *beyond* the training data and have the model transfer skills to new settings. To address the challenges, we leverage recent advances in **diffusion-model** large-scale text-to-image generation. Such models [33, 30, 29] can inherently model multimodality via their iterative denoising process. To effectively transfer skills to new settings, we propose a modified diffusion model with the ability to **discretize** learned behavior from language-annotated data. Figure 2 shows an overview of our method.

### 4.1   Language Conditioned Play Data

Our setup consists of language conditioned play data [21] $D_{\texttt{play}} = \{(s_t^{(i)}, a_t^{(i)})\}_{i=1}^N$: long sequences of robot behavior data containing many kinds of behaviors, collected by human operators instructed to perform interesting tasks. In this setting, we assume that there is some optimality to the data, i.e. $a_t \sim \mathcal{F}(s_t, z_g)$, where $z_g$ is a latent variable that models the intention of the operator. We thus leverage *language* labels to estimate $z_g$. Given a sequence $\tau = \{s_i, a_i\}_{t=k}^H$, label $\tau$ with an instruction $l$ which is passed into a language model [65], $g_{\text{lang}}$, referring to it as $z_l$ throughout the paper. One can also use goal images, but we might not have access to these at test time. While our method can use any $z_g$ as conditioning, assume that the play data has access to language annotations $l$. Our policy $\pi(a_t|s_t, z_l)$ contains a few simple components. We use a ResNet[66]-based visual encoder $\phi_v$ to encode $s_t$ (a sequence of images) and an MLP based langauge encoder $\phi_l$ to downproject the language embedding $z_l$. The policy uses $g = [\phi_l(z_l), \phi_v(s_t)]$ as conditioning to the action decoder $f_{\text{act}}$. Previous approaches [21, 34] use latent variable models to deal with multimodality. We find that modelling $f_{\text{act}}$ as a diffusion process enables us to circumvent this.

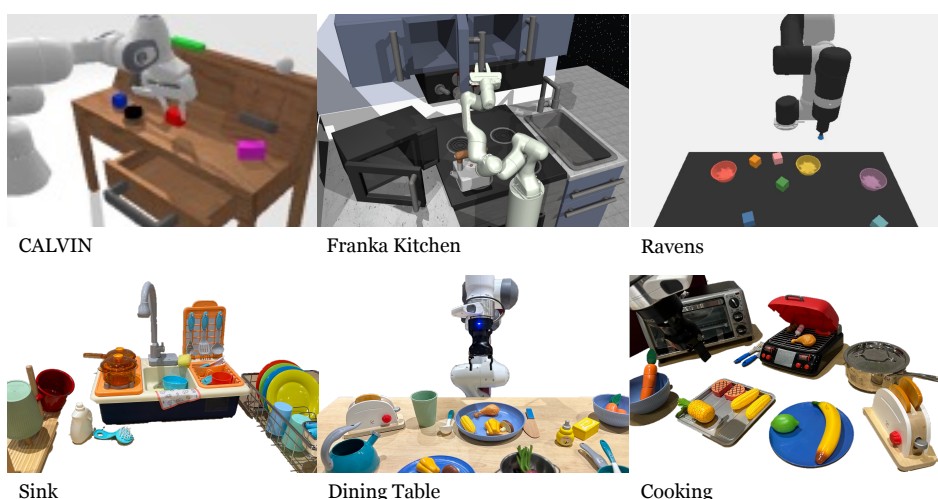

Figure 3: Simulated (top row) and real-world (bottom row) environments used for our evaluations. In each real-world setup, the robot is tasked with picking up one of the objects (e.g., plate, cup, carrot, bread, corn) and relocating it to a specified location (e.g., drying rack, plate, toaster, grill, pot).

## 4.2 Multi-modal Behavior Generation via Diffusion

With $f_{\text{act}}$, we aim to predict robot actions given the current state, using a DDPM to approximate the conditional distribution $P(a_t|s_t)$. In our setting, we additionally condition on the goal $g$. Formally, we train the model to generate robot actions $a_t$ conditioned on goal $g$ and current state $s_t$, so we modify Equations 1 and 6 to obtain:

$$a_t^{k-1} = \alpha(a_t^k - \gamma\epsilon_\theta(g, s_t, a_t^k, k) + N(0, \sigma^2 I)) \tag{4}$$

$$\mathcal{L} = ||\epsilon^k - \epsilon_\theta(g, s_t, a_t^0 + \epsilon^k, k)||_2 \tag{5}$$

We use the notation above for simplicity, but in practice, we predict a sequence of $T_a$ future actions $a_t, \cdots, a_{t+T_a}$ instead of only the most immediate action, a technique known as *action chunking*. This is done in some recent works [33, 67] and is shown to improve temporal consistency.

## 4.3 Discrete Diffusion for Control

Moreover, humans often break down tasks into smaller skills, which are often repeatable. In fact, most tasks can be achieved with a relatively small set. On the other hand, both the latent goals that we learn as well as the action diffusion process are continuous. Making sure learnt skills are discrete can not only allow for better performance but also better generalization to new settings. However, naively enforcing discretization can lead to suboptimal behavior. We want to ensure that conditioned on a latent goal, $g$, action predictions from $f_{\text{act}}$ are both multimodal and yet only represent a few modes. Thus, we propose a discrete *bottleneck* instead.

For the action generation process to represent a useful skill space, we want to enforce discreteness where the actions interact with latent goal. PlayFusion adds a vector quantization bottleneck in the diffusion process, specifically in the network $\epsilon_\theta(x) = \epsilon_\theta(g, s_t, a_t^0 + \epsilon^k, k)$. $\epsilon_\theta$ is U-Net which fuses the language conditioning into the action denoising. We modify the U-Net architecture with a codebook of discrete latent codes $e_u$, a discrete bottleneck for the diffusion model. Given an input $x$ the U-Net encoder produces a latent $\psi_\epsilon(x)$, which is passed into the decoder to produce $\epsilon_\theta(x) = \gamma_\epsilon(\psi_\epsilon(x))$. This bottleneck layer selects $j$ as $\arg\min_i ||\psi_\epsilon(x) - e_i||$, finding the closest code to the embedding, which is used to reconstruct $x$. To account for this, we augment the training procedure with the quantization and commitment losses, similar to VQ-VAE.

**Generalization via discrete language conditioning** Consider an agent that has learnt skills formed from the atomic units A, B, C and C, of the form A + B, B + C and C + D. To truly extend its

capabilities beyond the initial training data, the agent must learn to interpolate and extrapolate from these existing skills, being able to perform tasks like A + D that it hasn't explicitly been trained on. Given that the action generation in the diffusion process is already quantized, our hypothesis is that a discrete goal space will be synergestic and allow the policy to compose skills better. Thus, we maintain a codebook of discrete latent codes $e_l$ for the language embeddings output by the language goal network $\phi_l(z_l)$, selecting $e_{l,j}$ which is closest to $\phi_l(z_l)$. The full loss function that we use to train PlayFusion is as follows:

$$
\mathcal{L}_{PlayFusion} = ||\epsilon^k - \epsilon_\theta(x^0 + \epsilon^k, k)||_2 + \beta_1 \underbrace{||sg(\psi_\epsilon(x) - e_{u,j})||_2}_{\text{U-Net quantization loss}} + \beta_1 \underbrace{||\psi_\epsilon(x) - sg(e_{u,j})||_2}_{\text{U-Net commitment loss}}
$$
$$
+ \beta_2 \underbrace{||sg(\phi_l(z_l)) - e_{l,j}||_2}_{\text{lang. quantization loss}} + \beta_2 \underbrace{||\phi_l(z_l) - sg(e_{l,j})||_2}_{\text{lang. commitment loss}}
$$

(6)

where $\beta_1$ and $\beta_2$ are coefficients to determine the tradeoff between covering a diversity of possible behaviors and encouraging behaviors belonging to similar skills to be brought close to each other.

**Sampling from PlayFusion**    Given a novel language instruction at test time $z'$, we obtain the quantized encoding $\phi_l(z')$, combining it with the visual encoding to get conditioning $g'$. We sample a set of actions $a_{t:t+k} \sim \mathcal{N}(0,1)$, pass them through the discrete denoising process in Equation 4.

## 5   Experiments

In this section, we investigate PlayFusion and its ability to scale to complex tasks, as well as generalization to new settings. We ask the following questions: (1) Can PlayFusion allow for learning complex manipulation tasks from language annotated play data? (2) Can our method perform efficiently in the real-world setup beyond the simulated environment? (3) How well can PlayFusion generalize to out of distribution settings? (4) Can PlayFusion in fact learn discrete skills? (5) How do various design choices, such as quantization, language conditioning, etc., affect PlayFusion? We aim to answer these through experiments in three different simulation and real world settings.

**Environmental Setup**    We test our approach across a wide variety of environments in both simulations as well as the real world. For simulation, we evaluate three benchmarks: (a) CALVIN [34], (b) Franka Kitchen [22], and (c) Language-Conditioned Ravens [35, 36]. For the real-world setup, we create three different environments: cooking, dining table and sink, shown in Figure 3. More details of the environment setup are in the supplementary.

**Baselines**    We handle task conditioning in the same way for our method as well as all baselines, using the same visual and language encoders. We compare our method with the following baselines: (a) *Learning Motor Primitives from Play (Play-LMP)*: Play-LMP [21] generates behaviors by learning motor primitives from play data using a VAE, which encodes action sequences into latents and then decodes them into actions. (b) *Conditional Behavior Transformer (C-BeT)*: C-BeT [23] generates behaviors using a transformer-based policy and leverages action discretization to handle multimodality. (c) *Goal-Conditioned Behavior Cloning (GCBC)*: GCBC [21, 68] is conditional behavior cloning.

### 5.1   Results in Simulation and Real World

**PlayFusion in simulation**    Table 1 shows success rates for PlayFusion, Play-LMP, C-BeT, and GCBC on the simulation benchmarks. On both CALVIN setups, we outperform the baselines by a wide margin, which demonstrates the effectiveness of our method in large-scale language-conditioned policy learning from complex, multimodal play data. The baselines perform comparatively better on the Franka Kitchen environments, where the training datasets are smaller and the data covers a more narrow behavior distribution and the benefit of handling multimodality is smaller; however,

| | Simulation | | | | | Real World | | |
|---|---|---|---|---|---|---|---|---|
| | **CALVIN A** | **CALVIN B** | **Kitchen A** | **Kitchen B** | **Ravens** | **Dining Table** | **Cooking** | **Sink** |
| C-BeT [23] | $26.3 \pm 0.8$ | $23.4 \pm 0.9$ | $\mathbf{45.6 \pm 2.3}$ | $\mathbf{24.4 \pm 2.3}$ | 13.4 | 20.0 | 0.0 | 10.0 |
| Play-LMP [21] | $19.9 \pm 1.0$ | $22.0 \pm 0.4$ | $1.9 \pm 1.5$ | $0.0 \pm 0.0$ | 0.2 | 0.0 | 0.0 | 0.0 |
| GCBC [21] | $23.2 \pm 2.0$ | $30.4 \pm 1.4$ | $38.0 \pm 3.3$ | $15.5 \pm 4.5$ | 1.6 | 5.0 | 0.0 | 5.0 |
| Ours | $\mathbf{45.2 \pm 1.2}$ | $\mathbf{58.7 \pm 0.7}$ | $47.5 \pm 2.0$ | $\mathbf{27.7 \pm 0.9}$ | **35.8** | **45.0** | **30.0** | **20.0** |

Table 1: Success rates for PlayFusion and the baselines on simulation and real-world settings. PlayFusion consistently outperforms all of the baselines.

PlayFusion still outperforms or matches all baselines. PlayFusion also achieves significantly higher success rate than the baselines on Ravens (see appendix for per-task results), which is not as large-scale as CALVIN but covers a large portion of the state space due to the diversity of instructions.

**Long horizon tasks**  Using the Long Horizon CALVIN evaluation suite, we test the ability of agents to stitch together different tasks, with transitioning between tasks being particularly difficult. One such long horizon chain might be "turn on the led" → "open drawer" → "push the blue block" → "pick up the blue block" → "place in slider". We rollout 128 different long horizon chains containing five instructions each and record the number of instructions successfully completed. As shown in Table 2, we find that PlayFusion significantly outperforms the baselines in both CALVIN A and CALVIN B. The diffusion process gracefully handles the multimodality of not only each individual task in the chain but also of the highly varied data the agent has seen of transitions between tasks.

| | | No. of Instructions | | | | |
|---|---|---|---|---|---|---|
| | Av. Seq Len | 1 | 2 | 3 | 4 | 5 |
| *CALVIN A* : | | | | | | |
| C-BeT | 0.262 | 25.2 | 1.0 | 0.0 | 0.0 | 0.0 |
| Play-LMP | 0.175 | 16.5 | 1.0 | 0.0 | 0.0 | 0.0 |
| GCBC | 0.194 | 19.4 | 0.0 | 0.0 | 0.0 | 0.0 |
| *CALVIN B* : | | | | | | |
| C-BeT | 0.272 | 27.2 | 0.0 | 0.0 | 0.0 | 0.0 |
| Play-LMP | 0.117 | 11.7 | 0.0 | 0.0 | 0.0 | 0.0 |
| GCBC | 0.291 | 27.2 | 1.9 | 0.0 | 0.0 | 0.0 |
| Ours (A) | **0.417** | **37.1** | **2.9** | **1.0** | 0.0 | 0.0 |
| Ours (B) | **0.611** | **54.4** | **6.0** | 0.0 | 0.0 | 0.0 |

Table 2: Average sequence length on Long Horizon CALVIN and success rate for the $n$-th instructions.

**Generalization in the real world**  Table 1 shows results for PlayFusion and the baselines in our real world evaluation setups. These setups are particularly challenging for two reasons: (1) inherent challenges with real-world robotics such as noisier data and constantly changing environment conditions such as lighting, and (2) they are designed to test skill-level compositional generalization. Specifically, the agents are required to compose skills A + B and C + D into A + D; for example, they might be trained on "pick up the carrot and place it in the pan" and "pick up the bread and put it in the toaster" and must generalize to "pick up the carrot and put it in the toaster". Our method significantly outperforms the baselines in these settings, showcasing the ability of the diffusion model in modeling complex distributions and the emergence of learned skills via the discrete bottleneck. Video results are at `https://play-fusion.github.io`.

## 5.2  Analysis of Discrete Representations

**Learning discrete skills**  Table 3 studies the impact of our discrete bottlenecks (for Ravens results, see the appendix). The success rate is, on average, worsened with the removal of either the U-Net discretization and the language embedding discretization. We also qualitatively study whether semantically similar skills are actually mapped to similar areas of the latent space and should therefore be brought together by the discrete bottleneck. In Figure 4, we show that skills involving similar locations (e.g., pan)

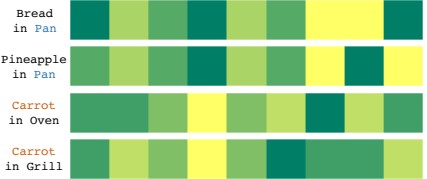

Figure 4: Visualization of the codebook embeddings for various real-world skills.

or objects (e.g., carrot) are encoded into similar embeddings. In Figure 4, we show the embeddings

of different trajectories. The top two rows share the first skill (which is to remove the lid from the pan) and place an object in the pan. The bottom two rows share the second skill (grasping the carrot). Embeddings that contain the same skill have a similar pattern, which further indicates that the latent skill space being learned is somewhat discretized.

**Balancing the discrete bottlenecks**   In Table 4, we study the effects of different $\beta_1$ and $\beta_2$ values on CALVIN A performance, i.e., the relative weightings for the additional terms in the loss function corresponding to the U-Net discretization and language embedding discretization. We find that $\beta_1 = \beta_2 = 0.5$ results in the best performance. In general, equally weighing the four additional losses (two for U-Net and two for language) leads to improved performance over imbalanced weightings. $\beta_1 = \beta_2 = 0.5$ is also better than $\beta_1 = \beta_2 = 1$, indicating that over-incentivizing discretization can be detrimental to diffusion model learning. Further analyses can be found in the appendix.

| Methods | CALVIN A | CALVIN B |
|---|---|---|
| Ours | **45.2 ± 1.2** | **58.7 ± 0.7** |
| No U-Net discretiz. | **45.3 ± 2.1** | 55.1 ± 1.4 |
| No lang discretiz. | 40.3 ± 1.6 | 54.1 ± 1.2 |

Table 3: Effect of discrete bottlenecks.

## 5.3   Ablations of Design Choices

**Effect of language model**   Although our method is orthogonal to the language model used, we test its sensitivity to this. As shown in Table 4, we find that common models such as MiniLM [65], Distil-roberta [69], MPNet [70], and BERT [71] have similar performance, showing that PlayFusion is mostly robust to this design choice. We hypothesize that the discrete bottleneck applied to the language embeddings helps to achieve this robustness. CLIP [72] embeddings result in much lower success rates, most likely due the fact that Internet images may not contain similar "play data" instructions.

**Effect of conditioning**   Table 4 studies various different possibilities for conditioning the diffusion model generations on language and vision in CALVIN A. When working with diffusion models there are multiple different ways we can approach how to feed it goals, images of the scene etc. We found that PlayFusion is mostly robust to this, with global conditioning providing benefits for smaller models (such as those in the real world). We also attempted to condition the diffusion model noise on the goal but found that this negatively impacted performance. For the visual conditioning, we studied the effect of initializing the image encoder with large-scale pre-trained models [73]), finding that it does not help, and PlayFusion can learn the visual encoder end-to-end from scratch.

|  | **Success Rate** |
|---|---|
| *Effect of conditioning*: | |
| Global | 54.1 |
| Conditional Noise | 40.2 |
| Visual Pre-training | 38.1 |
| *Effect of language model*: | |
| all-MiniLM-L6-v2 | 47.1 |
| all-distilroberta-v1 | 48.4 |
| all-mpnet-base-v2 | 48.8 |
| BERT | 48.8 |
| CLIP (ResNet50) | 35.2 |
| CLIP (ViTB32) | 43.9 |
| *Loss weights (U-Net & Language)* : | |
| 0.5 & 0.5 | 47.1 |
| 1 & 1 | 45.1 |
| 0.1 & 1 | 45.5 |
| 1 & 0.1 | 43.4 |
| 0.25 & 0.75 | 37.7 |
| 0.75 & 0.25 | 43.4 |

Table 4: Effects of conditioning, language model, and loss weights.

For data scaling curves and more analyses on design choices, see the appendix.

## 6   Limitations and Discussion

In this paper, we introduced a novel approach for learning a multi-task robotic control policy using a denoising diffusion process on trajectories, conditioned on language instructions. Our method exploits the effectiveness of diffusion models in handling multimodality and introduces two discrete bottlenecks in the diffusion model in order to incentivize the model to learn semantically meaningful skills. PlayFusion does require the collection of teleoperated play data paired with after-the-fact language annotations, which still require human effort despite being already less expensive and time-consuming to collect than demonstrations. It would be interesting to label the play data with a captioning model or other autonomous method. Furthermore, there is room for improvement in our performance on our real-world setups. Additionally, our real-world experiments could be expanded to even more complex household settings such as study rooms, bed rooms, and living rooms. Overall, our approach can significantly enhance the ability of robots to operate autonomously in complex and dynamic environments, making them more useful in a wide range of applications.

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

## A  Website

Video results are available at https://play-fusion.github.io.

## B  Experimental Setup

We evaluate our method on three simulated environments. Below, we provide their details.

**CALVIN [34].**  The CALVIN benchmark tests a robotic agent's ability to follow language instructions. CALVIN contains four manipulation environments, each of which include a desk with a sliding door and a drawer that can be opened and closed, as well as a 7-DOF Franka Emika Panda robot arm with a parallel gripper. The four environments differ from each other in both their spatial composition (e.g., positions of drawers, doors, and objects) and visual features. The training data for each environment contains around 200K trajectories, from which we sample a sequence of transitions for each element of the minibatch. A portion of the dataset contains language annotations; we use this subset to train our language-conditioned model. Each transition consists of the RGB image observation, proprioceptive state information, and the 7-dimensional action. The agent is evaluated on its success rate in completing 34 tasks, which include variations of rotation, sliding, open/close, and lifting. These are specified by language instructions that are unseen during training in order to test the generalization ability of the agent. We evaluate on two setups: (1) CALVIN A, where the model is trained and tested on the same environment (called D→D in the benchmark) and (2) CALVIN B, where the model is trained on three of the four environments and tested on the fourth (called ABC→D in the benchmark).

**Franka Kitchen [22].**  Franka Kitchen is a simulated kitchen environment with a Franka Panda robot. It contains seven possible tasks: opening a sliding cabinet, opening a hinge cabinet, sliding a kettle, turning on a switch, turning on the bottom burner, turning on the top burner, and opening a microwave door. The dataset contains 566 VR demonstrations of humans performing four of the seven tasks in sequence. Each transition consists of the RGB image observation, proprioceptive state information, and the 9-dimensional action. We split each of these demonstrations into their four tasks and annotate them with diverse natural language to create a language-annotated play dataset. In our experiments, we evaluate agents on two setups within this environment, which we denote as Kitchen A and Kitchen B. In Kitchen A, we evaluate an agent's language generalization ability at test-time by prompting it with unseen instructions asking it to perform one of the seven tasks. This requires the model to identify the desired task and successfully execute it. Kitchen B is a more challenging evaluation setting, where the agent must perform two of the desired seven tasks in sequence given an unseen language instruction. In this setting, the agent must exhibit long-horizon reasoning capabilities and perform temporally consistent actions, in addition to the language generalization required in Kitchen A.

**Language-Conditioned Ravens [35, 36].**  Ravens is a tabletop manipulation environment with a Franka Panda arm. We evaluate on three tasks in the Ravens benchmark: putting blocks in bowls, stacking blocks to form a pyramid, and packing blocks into boxes. The dataset consists of 1000 demonstrations collected by an expert policy. Although the dataset proposed in [36] contains language instructions denoting which color block to move and the desired final location, they are not diverse like human natural language annotations would be. In order to study our model's performance on a play-like language-annotated dataset, we instead annotate the demonstrations with diverse natural language. At test-time, we prompt the agent with an unseen language instruction, similar to our other setups.

### B.1  Real World Setup

We create multiple play environments in the real world as well. We use a 7-DOF Franka Emika Panda robot arm with a parallel gripper, operating in joint action space. We have three different

environments `cooking`, `dining table` and `sink`. All of these tasks are multi-step, i.e., in each the robot has to at least grab one object and put it in another, i.e. grab a carrot and put it inside the oven. In `cooking`, we test how the robot can handle articulated objects. It has to first open the oven, grill or pot, and then place an object properly inside. All of these objects have different articulations. Each of the placed objects (bread, carrot, knife, steak, spoon, etc.) have unique and different ways of being interacted with. In the `sink`, we test very precise manipulation skills, where the robot has to place objects in the narrow dish rack or hang objects (like mugs). In all of these settings, we test unseen goals (a combination of objects) that has never been seen before, as well as an instruction that has never been seen before. We provide more details in the Appendix.

## B.2 Additional Analysis on Discretization Bottleneck

**Discretization ablation in Ravens.** Table 5 studies the impact of our discrete bottlenecks on the Ravens benchmark. The success rate is, on average, worsened with the removal of either the U-Net discretization and the language embedding discretization.

**Discretizing a portion of the latent.** It is possible to quantize only a portion of the U-Net latent representation. Table 6 shows results of discretizing only a portion (25% or 50%) of the latent. We find that discretizing 25% of the latent resulted in better performance. Discretizing the entire latent still works well, but discretizing a portion is a great way to balance encouraging skill learning and accurate denoising.

Table 5: Effect of discrete bottlenecks on Ravens tasks.

| Methods | put-block-in-bowl | stack-block-pyramid | packing-box-pairs |
|---|---|---|---|
| Ours | **63.6 ± 2.5** | **20.0 ± 0.0** | **24.0 ± 1.8** |
| No U-Net discretization | **65.5 ± 3.3** | 5.0 ± 2.3 | 18.5 ± 0.0 |
| No lang discretization | 4.1 ± 0.6 | 3.3 ± 2.7 | 7.5 ± 2.5 |

Table 6: Effect of discretizing different fractions of the U-Net representation.

| Methods | Success Rate |
|---|---|
| Discretize 100% of latent | 45.2 ± 1.2 |
| Discretize 50% of latent | 44.8 ± 0.1 |
| Discretize 25% of latent | **48.7 ± 0.8** |

## B.3 Data Scaling Curves

Figure 5 shows data scaling curves.

**Effect of discrete bottlenecks.** Our method scales well with more data and performs very well even at 100K trajectories, which is half the size of the CALVIN A training dataset. The removal of the language discretization results in lower success rates across almost all dataset sizes. The removal of U-Net discretization is not as critical and can actually improve performance for very small datasets, but is on average harmful for larger datasets.

**Comparison to baselines.** Our method scales well with more data while C-BeT, Play-LMP, and GCBC perform poorly for all dataset sizes.

## B.4 Dataset Details

**Real-world experiments.** For each environment we collected 250 episodes. This translates to around 15 hours of data collection. We augmented the dataset by adding 3 or 4 variations for each language

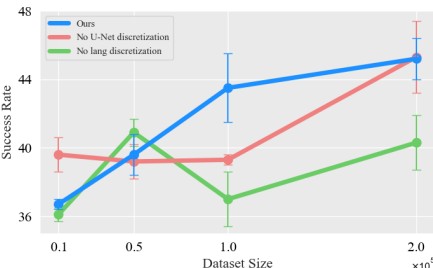 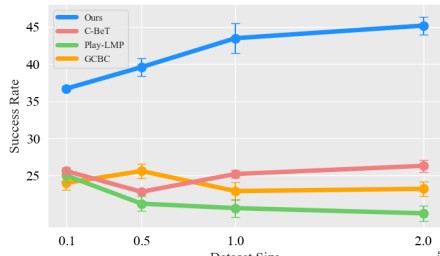

Figure 5: Data scaling curves. Left: effect of discrete bottlenecks. Right: comparison to baselines.

instruction (making the training dataset 750-1K episodes). The episodes were not broken into smaller annotated instructions.

**Simulation experiments.** We directly use the language-annotated dataset from CALVIN [34] and data generation script from CLIPort [36]. For Kitchen experiments, we used the dataset from Relay Policy Learning [22] and performed some processing and annotation to create language-annotated datasets. We provide some information in Table 7, but note that some of the numbers are estimates due to data processing procedures and refer the reader to the papers [34, 36, 22] for full details.

Table 7: Dataset details for simulation experiments.

|  | How was play data collected? | Hours | Eps. length | No. of lang. annotated eps. | Is a single eps. broken into smaller instructions? |
|---|---|---|---|---|---|
| CALVIN A | Teleoperators are instructed only to explore. Processing into episodes and annotating with language are done after-the-fact. | 2.5 | 64 | 5K (instructions are repeated to create 200K training episodes) | No. Training trajectories are length-16 sub-episodes of the length-64 episodes. Instructions are repeated for all sub-episodes to create a total of 200K language-annotated training trajectories. (However, the length-64 window was sampled from a long stream of play data). |
| CALVIN B | Same as CALVIN A, but for three different environments. | 7.5 | 64 | 15K (instructions are repeated to create 600K training episodes) | No. Same as CALVIN A, but for three different environments, for a total of 600K training trajectories. |
| Kitchen A | Teleoperators are instructed to perform 4 out of 7 possible tasks for each episode. | 1.5 | 200 | 566 (split to create 2.2K training episodes) | Yes. We split each episode into the four training trajectories and label each of them with language. |
| Kitchen B | Same as Kitchen A. | 1.5 | 200 | 566 (split to create 1.6K training episodes) | Yes. We split each episode into three training trajectories (one for each pair of consecutive tasks) and label each of them with language. |
| Ravens | Data is generated by rolling out an expert policy. | 3 | Up to 20 | 1000 | Depends on the task. If it is sequential then the instruction changes throughout the episode and if it is single-step then there is one instruction for the episode. |

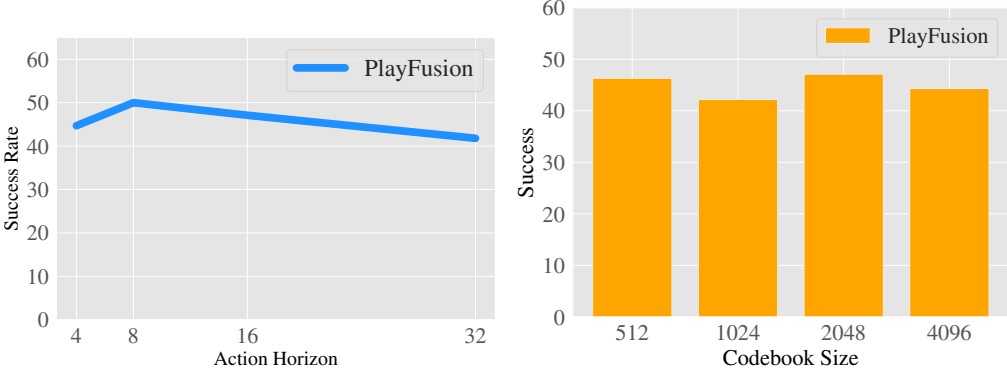

Figure 6: Effect of model design choices.

## B.5   Model Design Choices

Figure 6 studies the impact of action horizon and codebook size in CALVIN A. PlayFusion is mostly robust to the action horizon $T_a$. We empirically found $t_a$ of around 20% of the overall horizon worked the best. We find that PlayFusion is relatively robust to the discrete latent codebook sizes.

Note that asymptotically, increasing the codebook size would remove the discrete bottleneck, in principle. To study whether this happens in practice, we further increased the codebook size and show CALVIN A results in Table 8. As expected, performance drops when codebook size gets very large.

Table 9 shows the effect of number of diffusion timesteps in CALVIN A. We found that using 25 timesteps works slightly better but our method is generally robust to this hyperparameter.

Table 8: Effect of further increasing the codebook size.

| Codebook Size | Success Rate |
| --- | --- |
| 2048 | **45.2 $\pm$ 1.2** |
| 8192 | **46.0 $\pm$ 1.2** |
| 16384 | 41.1 $\pm$ 0.1 |

Table 9: Effect of diffusion timesteps.

| Timesteps | Success Rate |
| --- | --- |
| 50 | 45.2 $\pm$ 1.2 |
| 100 | 39.9 $\pm$ 1.3 |
| 25 | **47.4 $\pm$ 0.8** |

## B.6   Generalization to Unseen Skills

We performed an experiment where we removed one skill from the CALVIN training data. Specifically, we removed lift-red-block-slider from the training data and tested the model's ability to interpolate between (1) lifting other blocks from the slider (e.g., lift-blue-block-slider, lift-pink-block-slider) and (2) lifting red blocks in other scenarios (e.g., lift-red-block-drawer, lift-red-block-table). We also repeated this experiment for lift-blue-block-table. We find that the removal of the discrete bottlenecks results in generally worse performance in this challenging setup (see Table 10). Although confidence intervals do overlap a bit, we find that our method is on average the best for both lift-red-block-slider and lift-blue-block-table.

Table 10: Performance on unseen skills.

| Models | lift-red-block-slider | lift-blue-block-table |
|---|---|---|
| Ours | $20.0 \pm 8.1$ | $16.6 \pm 7.2$ |
| No U-Net discretization | $10.0 \pm 4.6$ | $3.3 \pm 2.7$ |
| No lang discretization | $13.3 \pm 2.7$ | $13.3 \pm 5.4$ |

## B.7 Ravens Experiments

Table 11 shows per-task success rates for Ravens.

Table 11: Per-task success rates for Ravens.

| | put-block-in-bowl | stack-block-pyramid | packing-box-pairs |
|---|---|---|---|
| C-BeT | $17.2 \pm 1.1$ | $15.0 \pm 2.3$ | $8.1 \pm 1.5$ |
| Play-LMP | $0.0 \pm 0.0$ | $0.0 \pm 0.0$ | $0.8 \pm 0.2$ |
| GCBC | $0.0 \pm 0.0$ | $3.3 \pm 2.7$ | $1.7 \pm 0.7$ |
| Ours | $\mathbf{63.6 \pm 2.5}$ | $\mathbf{20.0 \pm 0.0}$ | $\mathbf{24.0 \pm 1.8}$ |

## B.8 Implementation Details

Table 12 shows the main hyperparameters of our model in our simulation and real world experiments. We build off of the implementation of MCIL from CALVIN [74]. For Franka Kitchen and Ravens dataset and environment processing, we use implementations from [75] and [76], respectively. For implementations of the baselines, we modify [77] for C-BeT and [74] for Play-LMP and GCBC. Where possible, we use the same hyperparameters for PlayFusion and the baselines.

Table 12: Hyperparameters of PlayFusion in our simulation and real-world experiments.

| Hyperparameter | CALVIN | Franka Kitchen | Ravens | Real World |
|---|---|---|---|---|
| Batch size | 32 | 32 | 128 | 12 |
| Codebook size | 2048 | 2048 | 2048 | 2048 |
| U-Net discretiz. wgt | 0.5 | 0.5 | 0.5 | 0.5 |
| Lang. discretiz. wgt | 0.5 | 0.5 | 0.5 | 0.5 |
| Action horizon $T_a$ | 16 | 64 | 2 | 32 |
| Context length $T_o$ | 2 | 1 | 1 | 1 |
| Language features | 384 | 384 | 384 | 384 |
| Learning rate | 1e-4 | 2.5e-4 | 2.5e-4 | 2.5e-4 |
| Diffusion timsteps | 50 | 50 | 50 | 50 |
| Beta scheduler | squaredcos_cap_v2 | squaredcos_cap_v2 | squaredcos_cap_v2 | squaredcos_cap_v2 |
| Timestep embed dim | 256 | 256 | 128 | 256 |

