# OpenReview forum: "PlayFusion: Skill Acquisition via Diffusion from Language-Annotated Play"
_robot-learning.org/CoRL/2023/Workshop/TGR — CoRL 2023 Workshop TGR Poster_

### Official Review · Reviewer_eCKL · 2023-10-19

**Rating:** 7
**Confidence:** 3

**Review:**

This paper aims to learn goal-directed skill policies from plays (unstructured) data. The authors propose PlayFusion, a diffusion model which can learn from language-annotated play data via discrete bottlenecks. Such an approach for skill acquisition can be highly relevant with the topic of this workshop about learning toward generalist robot.

---

### Official Review · Reviewer_rdYQ · 2023-10-20

**Rating:** 9
**Confidence:** 4

**Review:**

Despite challenges posed by the data collection process, this paper introduces a novel approach to distilling robot skills from play data combined with language annotations. It further explores the process of achieving new skills through interpolation and extrapolation.

---

### Decision · Program_Chairs · 2023-10-20

**Decision:**

Accept (Poster)

**Comment:**

Great paper and closely aligned topic!